# Differences in the Temporal Kinetics of the Metabolic Responses to Salinity Between the Salt-Tolerant *Thellungiella salsuginea* and the Salt-Sensitive *Arabidopsis thaliana* Reveal New Insights in Salt Tolerance Mechanisms

**DOI:** 10.3390/ijms26115141

**Published:** 2025-05-27

**Authors:** Aayush Sharma, Tahar Taybi

**Affiliations:** School of Natural and Environmental Sciences, Newcastle University, Newcastle upon Tyne NE1 7RU, UK

**Keywords:** salt tolerance, *Arabidopsis thaliana*, *Thellungiella salsuginea* (*halophila*), sugars, proline, malate

## Abstract

To unravel the mechanisms underpinning salt tolerance, different studies have attempted to determine the physiological and genetic variations behind the difference in salt tolerance between *Arabidopsis thaliana* and the salt-tolerant *Thellungiella salsuginea* (*halophila*). Most of these studies were limited to a specific duration of salt treatment and neglected the time response as a possible contributing factor to the higher salt tolerance exhibited by *T. salsuginea*. In this work, a comprehensive and detailed comparison of the response of the two species to high salinity was conducted at different times for up to ten days of salt treatment. *T. salsuginea* responded more rapidly and to a higher extent to adjust its metabolism and showed constitutive levels of anticipatory metabolism to salinity. *T. salsuginea* responded more rapidly in terms of maintaining light use efficiency, limiting the uptake of Na^+^, and increasing the accumulation of sugars and proline when exposed to salinity. *T. salsuginea* had much higher constitutive levels of metabolites, including malate, proline, and inositol, in comparison to *A. thaliana*. Interestingly, *T. salsuginea* showed a reduction in malate levels under salt treatment in contrast to *A. thaliana*. These results suggest that constitutive levels and the rapidity of the deployment of resistance mechanisms, together with metabolic plasticity, in response to salt stress are important adaptive traits for salt tolerance in plants.

## 1. Introduction

The hypotheses being tested in this work were that (1) the difference in salt tolerance between *A. thaliana* and *T. salsuginea* lies in part in the differential regulation of same response mechanisms exhibited by both species and (2) the degree of metabolic plasticity and rapidity and the amplitude of the response to salt stress are important factors in determining the level of salt tolerance. Salinity is a major abiotic stress that causes large reductions in plant growth and crop yield in many areas of the globe. Large variations in salt tolerance exist among plants from different species and in some cases between varieties of the same species [1]. While it is accepted that similar mechanisms might be deployed in response to salinity in salt-tolerant and salt-sensitive plants, it is still not clear how plants adapted to salinity can better use these mechanisms to survive and grow under high salt concentrations [2,3]. Plants deploy different types of responses to salinity to ensure osmotic adjustment and ionic homeostasis and to prevent cellular toxicity caused by high salt concentrations in the cell [4,5]. Usually, the closure of stomata to limit water loss and the reduction in cell expansion in the roots and young leaves are rapidly deployed after exposure to salinity. Na^+^ exclusion by roots might be engaged to reduce the accumulation of Na^+^ ions within leaves; however, under high salinity, Na^+^ and/or in some cases Cl^−^ ions accumulate in the plant and might be sequestered in the vacuole [2]. Consequently, compatible solutes accumulate proportionally to change the osmotic potential of the cell to balance the osmotic pressure exerted by the ions in the vacuole [6,7]. Usually, accumulated osmolytes include sucrose and fructose (sugars), methylated inositols, and glycerol (sugar alcohols), as well as trehalose and fructans as complex sugars [8]. Along with these, other osmolytes include charged ions like K^+^ or charged compounds like Dimethylsulfopropionate (DMSP) and glycine betaine, ectoine, and proline [9]. Metabolites like sucrose, glycine betaine, and proline are commonly the most accumulated compounds. However, certain species can accumulate other species-specific compounds [9,10,11]. Under high salt concentrations, compatible solutes not only lower the inner osmotic potential of the cell to facilitate the osmotic adjustment but also act as osmoprotectants [12,13]. As these solutes are hydrophilic in nature, they can easily take the place of water in cells and protect proteins and protein complexes as well as membranes [9]. Compatible solutes can limit the inhibition of enzyme activities caused by sodium ions and increase enzymes’ thermal stability, avoiding complex dissociation [9]. The synthesis of compatible solutes is connected to different metabolic pathways and thus can impact cellular metabolism [9]. The synthesis of compatible solutes involves the use of carbon and energy consumption, leading to a low plant growth rate. While only three and half ATPs are required to accumulate one Na^+^, forty-one ATP molecules are required to make one proline and around fifty-two ATP molecules are needed to make one molecule of sucrose [14]. This suggests that a plant must be prepared to mobilize a lot of energy and carbon to escape and/or recover from salt stress [6]. This requires high metabolic plasticity in terms of carbon and energy allocation, which might be a trait of salt-tolerant plants.

Work has been conducted in glycophytes and halophytes to understand the role of single metabolites or groups of metabolites and/or compatible solutes in salt tolerance. *A. thaliana* and *T. salsuginea* have largely contributed to the knowledge we possess today about changes in metabolites in general under salinity. In [15], the authors profiled metabolic changes in *A. thaliana* and *T. salsuginea* to look for common and divergent responses to salt treatment in the two species. Their work has shown differential metabolic responses between *A. thaliana* and *T. salsuginea* and that these changes might be mediated in *A. thaliana* by changes in abscisic acid and jasmonic acid levels. In contrast, in *T. salsuginea*, minimal changes in hormonal levels were induced by salinity, while more changes in metabolite levels took place in this species compared to *A. thaliana* [15]. Differential metabolic responses to salinity may take place in varieties of the same species. In salt-tolerant varieties of barley, salinity caused increased levels of sugar phosphates, intermediates of the TCA cycle, and metabolites involved in cellular protection (e.g., antioxidants). In contrast, in sensitive varieties, salinity increased the levels of amino acids, including proline, which was interpreted as a consequence of reduced growth [16].

Different studies have compared *A. thaliana* and *T. salsuginea* in the past, providing important insights into the possible mechanisms underpinning the differential salt tolerance exhibited by the two species. However, none of these studies have considered the kinetics of the response to salinity in terms of the rapidity and amplitude of the deployment of the salt-tolerance mechanisms as putative elements governing or participating in the higher salt tolerance exhibited by *T. salsuginea*. In this work, we conducted a detailed comparative analysis of the responses of *A. thaliana* and *T. salsuginea* to salinity over a 10-day course. The growth response and maximum light use efficiency (maximum efficiency of PSII) were compared in *A. thaliana* and *T. salsuginea* under salt treatment. Also, the accumulation of Na^+^ and K^+^ was compared in the two species under salt treatment. The regulation of the uptake and compartmentalization of Na^+^ is a key aspect in salt tolerance. The uptake of sodium can impact on the uptake of K^+^ (a key mineral nutrient), in turn impacting on the growth and fitness of the plant. We also measured the differential accumulation of key metabolites, including sucrose, fructose, glucose, inositol, proline, and malate, in the two species subjected to a time course of salt treatment at two different NaCl concentrations. The accumulation of sugars, proline, and organic acids under salt stress plays major roles in osmoregulation and antioxidant responses in many plants. Therefore, it was crucial to compare the differential levels of these metabolites in the shoots of control and salt-treated plants of *A. thaliana* and *T. salsuginea*. We also monitored the change in transcript levels for the genes encoding Δ^1^-Pyrroline-5-carboxylate synthase 1 (*P5CS1*), a key enzyme in the production of proline, salt overly sensitive (*SOS1*), a salt transporter, and sucrose synthase 3 (*SUS3*), a key enzyme in the production of sucrose, to see if salt-induced changes in metabolite levels within and between species were controlled at the gene level.

## 2. Results

### 2.1. Effect of Salinity on Growth of A. Thaliana and T. salsuginea

As shown in Figure 1, salinity caused adverse effects on the growth of both *A. thaliana* and *T. salsuginea*. Under 100 mM [NaCl], the loss in biomass accumulation was greater in *A. thaliana* than in *T. slsuginea*. The onset of growth reduction began early in *A. thaliana* within the first week of salt treatment at both 50 and 100 mM [NaCl]. In *T. salsuginea*, the impact of salt treatment on growth became apparent after only 2 weeks. The dry weight accumulated in *A. thaliana* at 50 mM [NaCl] was less than half that of control plants, and at 100 mM NaCl, it was less than a third of that of the control plants after 4 weeks of salt treatment. In *T. salsuginea*, the dry weight accumulated after 4 weeks of salt treatment at 100 mM [NaCl] was about half that accumulated in control plants. *T. salsuginea* plants completely stopped growth at 500 mM [NaCl]; however, the leaves remained green.

### 2.2. Effect of Salinity on Photosynthesis in A. thaliana and T. salsuginea

Overall, salt treatment resulted in a substantial increase in background chlorophyll fluorescence. After 10 days, Fo, which represents the minimal chlorophyll fluorescence, showed a substantial increase in both *A. thaliana* and *T. salsuginea* under salt treatment at 100 mM NaCl for *A. thaliana* and 500 mM [NaCl] for *T. salsuginea*, indicating a reduction in light absorption capacities. This increase in Fo followed a transient decrease in the first 24 h of salt treatment in the two plant species (Figure 2). In *A. thaliana*, there was initially a slight decrease in Fo during the first 12 h, followed by an increase in both the salt-treated plants and the control plants. Fv/Fm, which measures the maximum quantum yield of photosystem II, showed reciprocal changes to those observed for Fo, indicating a reduced PSII efficiency in *A. thaliana* and *T. salsuginea* at 100 mM and 500 mM [NaCl], respectively. There was no important difference in Fv/Fm values between the control plants and plants treated with 50 mM [NaCl] for *A. thaliana* (Figure 2), whereas in *T. salsuginea* plants subjected to 500 mM [NaCl], a substantial decline in Fv/Fm was apparent after 24 h of salinity (Figure 2). When Fv/Fm was compared at 100 mM [NaCl] between the two species, *A. thaliana* was more affected after 3 days of exposure to salinity.

### 2.3. Effect of Salinity on Ion Uptake: Na^+^ and K^+^

As shown in Figure 3, both species showed increased accumulation of sodium ions under salt treatment. *A. thaliana* accumulated sodium at similar levels under both 50 and 100 mM [NaCl] treatments. After 10 days of salt treatment, the levels of Na^+^ in the shoots reached about 6-fold the level in the control plants. In *T. salsuginea*, there was, however, under 100 mM NaCl, a rapid transient increase in Na^+^ content after 1 day of exposure to salt, followed by a decline in the Na^+^ level in the shoots and finally a slight increase after 5 days of salt treatment. *T. salsuginea* plants exposed to 500 mM [NaCl] had a restricted Na^+^ uptake during the first day of salt treatment; however, a consistent increase in Na^+^ content followed, with Na^+^ levels reaching over 10 times those in the unstressed plants after 10 days of salt treatment. Although there was a general decrease in the concentration of K^+^, including in the control plants for both *A. thaliana* and *T. salsuginea*, interestingly, the pattern of change in K^+^ levels was different in the two species. In *T. salsuginea*, there was first an increase in K^+^ levels during the first day of salinity, particularly under 500 mM [NaCl], followed by a decrease (Figure 3).

### 2.4. Effect of Salinity on Accumulation of Sugars and Sugar Alcohols

There was an increase in sugar and inositol levels under salt treatment compared to the untreated controls in both plant species, with a greater increase seen in *T. salsuginea* (see Figure 4 and Figure 5). At 100 mM NaCl, there was about 4-fold and 2-fold more fructose and glucose, respectively, in *T. salsuginea* compared to the control, although there was 2-fold more sucrose in *A. thaliana* (Figure 5). The basic levels of sucrose and glucose at T0 in *T. salsuginea* were around twofold higher than those in *A. thaliana*. Moreover, the accumulation of glucose in *A. thaliana* decreased under salt treatment. On the other hand, *T. salsuginea* showed a 4-fold increase in sucrose accumulation under high salinity compared to the controls after 10 days of salt treatment. *T. salsuginea* maintained higher fructose and glucose levels in both the absence and presence of salt stress compared to *A. thaliana*. There was a substantial increase in inositol in both plant species under salt treatment. The results show a strong correlation between inositol accumulation and salt tolerance. *T. salsuginea* maintained higher levels of inositol in the presence and absence of salt (Figure 5).

### 2.5. Effect of Salinity on Proline Accumulation

As shown in Figure 6, large increases in the levels of proline were induced by salt treatment in both *A. thaliana* and *T. salsuginea*. The levels of proline in control plants of *T. salsuginea* were twice the level of those in control plants of *A. thaliana.* The pattern of the increase in proline levels in the two plant species under salt treatment was different. While there was a continuous slow accumulation of proline in *A. thaliana*, there was a strong and rapid increase in the proline levels in *T. salsuginea*, reaching a maximum after 3 days of salt treatment followed by a significant decline. Although there was no difference in the levels of proline in *T. salsuginea* subjected to100 and 500 mM [NaCl], there was a significant difference in those measured in *A. thaliana* plants subjected to 50 and 100 mM [NaCl]. It is worth noting that the levels of proline in *A. thaliana* and *T. salsuginea* were similar after 10 days of salt treatment.

### 2.6. Effect of Salinity on Malate Accumulation

As shown in Figure 7, salinity had contrasting effects on the change in malate content between the two plant species. While there was a small increase in malate in *A. thaliana* under salt stress, there was a significant decrease in the amounts of malate over the 10 days of salt treatment in *T. salsuginea*. However, even after this decrease, *T. salsuginea* maintained higher levels of malate under the presence and absence of salt compared to *A. thaliana*. There was a substantial difference between malate levels in the unstressed plants of the two species. *T. salsuginea* had 4 times more malate than *A. thaliana* (Figure 7).

### 2.7. Effect of Salinity on Transcript Levels for P5CS1, SOS1, and SUS1 Genes in Shoots of A. thaliana and T. salsuginea

The differential regulation of metabolic activities between *A. thaliana* and *T. salsuginea* under salt treatment was investigated at the gene level. The transcript levels of the *P5CS1* gene were monitored in *A. thaliana* and *T. salsuginea* in control plants and plants subjected to salt treatment for up to 10 days and expressed relative to the controls. As shown in Figure 8, different kinetic profiles for the *P5CS1* transcript abundance in shoots were observed between the two species. Salt treatment caused a more rapid and higher increase in *P5CS1* transcripts in *T. salsuginea* than in *A. thaliana*. This change was mirrored by the change in the accumulation of proline in shoots (Figure 6). *A. thaliana* showed a maximum of three-fold more transcripts than the control under 100 mM [NaCl] treatment after 5 days, with a drop to one-fold after 10 days of salt treatment. In contrast, *T. salsuginea* rapidly accumulated *P5CS1* transcripts, reaching 5 times the control level after the first 3 days of salt treatment at 100 mM and 500 mM [NaCl], followed by a decline in transcript levels. There was no dose effect of salt on the *P5CS1* transcript levels in *T. salsuginea*, whereas treatment with 100 mM [NaCl] resulted in higher transcript levels in the shoots of *A. thaliana* at 5 days of salt treatment.

Figure 8C,D show changes in *SOS1* transcript levels in the shoots and roots of *A. thaliana* and *T. salsuginea* under the effect of salt treatment over a period of 10 days. There was no significant change in relative transcript levels for the *SOS1* gene in the shoots of either plant species under salt treatment at 50 and 100 mM [NaCl]. In *T. halophila*, salt treatment at 500 mM [NaCl] resulted in a transient, approximately three-fold increase in transcript levels compared to the control at 24 h of salt treatment. *SOS1* relative transcript levels declined slightly in the roots during the first 24 h of salt treatment. This decrease was transient, and a steady increase in *SOS1* relative transcript levels was induced by salt treatment in both plant species after one day of salt treatment. This increase continued in *A. thaliana* for up to 10 days of salt treatment and was stronger at 50 mM [NaCl] and reached over twice the T0 value. In *T. salsuginea*, *SOS1* relative transcript levels reached five times those at T0 in the roots of plants treated with 500 mM [NaCl] after 5 days and were over two-fold higher at 100 mM [NaCl]. Over the next 5 days, the relative transcript levels at 500 mM [NaCl] came down, reaching same levels as at 100 mM [NaCl], which were still higher than the levels at T0.

The transcript levels of a gene encoding sucrose synthase 3 (*SUS3*, At4g02280) were determined. As shown in Figure 8E,F, at 100 mM [NaCl], *A. thaliana* showed a minor increase in relative SUS transcript levels on the first day and fifth day of salt treatment to be about 1.5-fold more than those of control plants, before reducing by the tenth day of salt treatment. On the other hand, in *T. salsuginea*, SUS transcript levels became around 1.7-fold higher, increasing slowly until the end of the salt treatment on the 10th day. But *T. salsuginea* exhibited a dose–response, as at the higher [NaCl] level of 500 mM, the transcript level increased rapidly to be 1.7-fold and then 3-fold higher than T0 after 12 h and 3 days, respectively.

As shown in Table 1 below, there was a positive correlation between the measured change in transcript levels for the *P5CS1* gene and accumulated proline during the 10-day course of salt stress at 100mM [NaCl] in both *A. thaliana* and *T. Salsuginea*. There was, however, a positive correlation between the change in transcript levels for *SUS3* and accumulated sucrose only in *T. salsuginea* over the 10-day course of salt stress at 100 mM [NaCl].

## 3. Discussion

### 3.1. Impact of Salinity on Growth and Photosynthetic Capacities

As expected, salinity severely reduced the growth of *A. thaliana* and *T. salsuginea*, and the growth reduction was higher in *Arabidopsis* (*p* = 0.0002 and 0.0011 at 100 mM [NaCl], after 10 days, for *A. thaliana* and *T. salsuginea*, respectively). This reduction in growth might be the consequence of a drop in photosynthetic capacities and the deployment of stress resistance mechanisms, which are large sinks for energy and carbohydrates. The drop in photosynthetic capacities might be a consequence of the deployment of regulatory responses to limit the light energy harvested in the chloroplast to limit oxidative stress indirectly imposed by salinity on the photosynthetic machinery. Even under high salinity (500 mM [NaCl]), *T. salsuginea* seemed to limit oxidative stress by reducing photosynthesis and growth. This helps the plant to survive longer periods of stress and might lead to increased levels of photosynthesis and growth if conditions become more favorable.

### 3.2. Ion Selectivity: Na^+^ and K^+^ Content Under Salt Stress

*A. thaliana* and *T. salsuginea* demonstrated differences in the accumulation of sodium and potassium over the period of 10 days of salt treatment under two different concentrations of NaCl. Under 50 and 100 mM [NaCl], *A. thaliana* showed a gradual increase in the accumulation of Na^+^ rapidly after salt application, and this increase continued over 10 days of salt treatment (*p* = 0.0197). In contrast, *T. salsuginea* kept Na^+^ uptake under 100 mM [NaCl] throughout the 10 days at relatively low levels (*p* = 0.0001 at 100 mM [NaCl] for 10 days). This suggests that *T. salsuginea* might have a strong capability to restrict the uptake of Na^+^ by the roots and/or at the root/shoot barrier, allowing it to maintain low Na^+^ levels in the leaves. However, this feature is activated only at relatively moderate salt concentrations, e.g., 100 mM [NaCl], which is low for halophytes. At high salt concentrations (500 mM [NaCl]), *T. salsuginea* restricted salt uptake during the first day of salt treatment but accumulated large amounts of salt after that, suggesting that the control of the uptake of salt was ineffective beyond a certain concentration of salt in the soil, which explains the fact that growth under 500 mM [NaCl] was effectively nil. In glycophytes, excessive Na^+^ content is considered highly toxic and has harmful effects on plant growth and acts as a key destructive factor [17,18]. The authors of [19] suggested that an excessive amount of Na^+^ in tissue is the main factor behind the level of salt sensitivity of non-halophytes like *A. thaliana*; however, research conducted on Salt Overly Sensitive mutants of *A. thaliana* did not show any correlation between sodium content and salt sensitivity. *SOS1* mutants exhibited lower amounts of Na^+^ uptake but did not show any reduction in sensitivity to salt when compared to the wild type [18]. The amount of sodium taken up by the roots depends on the expression and activity of three types of sodium transporters located in the plasma membrane of root cortical and epidermal cells. These transporters include the high-affinity potassium transporters (HKTs), which are divided into two classes, class 1 (HKT1), which has greater affinity to Na^+^, and the HKT2 class, which transports either Na^+^ or K^+^; and the non-selective cation channels (NSCCs), which are capable of transporting Na^+^, K^+^, Ca^2+^, and Mg^2+^ into the root cells as well as the salt overlay sensitive *SOS1* transporter, which is specific to Na^+^ and acts to transport the ion out of the root cells [20]. *SOS1* plays a vital role in salt tolerance, the reduction in the expression of the *SOS1* gene in *T. salsuginea*, and the loss of function of the gene in *A. thaliana* that results in hypersensitivity to salt [21,22]. Interestingly, there was no significant change in the *SOS1* relative transcript levels, except for in *T. salsuginea* after one day of salt treatment at 500 mM [NaCl]. This did not correlate with the increase in Na^+^ levels in both species under salt treatment, which implies that the *SOS1* transporter is probably regulated by protein levels and/or post-translational modifications.

A much closer relationship was established between salt tolerance and the level of K^+^ content in the plant tissues [22]. The two species have shown different kinetics for the change in K^+^ levels under salt treatment. In *T. salsuginea*, there was first an increase in the K^+^ levels during the first 24h of salt treatment followed by a decrease, while in *A. thaliana*, exposure to 100 mM [NaCl] resulted in a decrease in K^+^ levels, followed by a transient recovery. However, the overall levels of K^+^ remained higher in *T. salsuginea* compared to *A. thaliana* over the 10-day period of salt treatment. It has been reported that *T. salsuginea* exhibits a higher preference for K^+^ over Na^+^ [23,24]. Potassium homeostasis plays an important role in many cellular processes, and small changes in the cellular K^+^ content can result in huge differences in plant growth rates [18]. The substantial decrease in K^+^ content observed at 500 mM [NaCl] in *T. salsuginea* can be explained by the direct competition exhibited by Na^+^ against root transporters. Chemically Na^+^ and K^+^ are very similar ions, and at very high concentrations, external Na^+^ has a limiting effect on K^+^ uptake by the roots [25,26]. Many different reports have suggested K^+^ is an important nutrient during salt stress in plant cell cultures and yeast [18]. Potassium is a very important element that plays critical roles not only in supporting plant growth but also in metabolism and various other cellular processes and is required by plants in large amounts. Therefore, the ability to take up and maintain high tissue content of K^+^ in the presence of excessive amounts of external or internal Na^+^ is a crucial factor in salt tolerance.

### 3.3. Impact of Salinity on Metabolite Accumulation

The metabolic responses of *A. thaliana* and *T. salsuginea* were compared over a time course to identify the difference in the accumulation kinetics of key metabolites that may participate in the differential salinity tolerance shown by the two species. The two species exhibited similar changes in selected metabolites but with differences in the kinetics and amplitude of change under salt treatment. The overall increases observed in sucrose, glucose, fructose, inositol, and proline showed a positive correlation with the salt concentration and increased to higher levels in *T. salsuginea* compared to *A. thaliana* (*p* < 0.005). This increase might be controlled at the gene level, since a differential change in the transcript levels of SUS1 similar to that observed for sugars was measured in the two plant species.

Soluble sugars such as sucrose, glucose, fructose, and inositol are the direct products of photosynthesis and components of primary metabolism. They may also result from the degradation of starch, which usually increases under stress [27]. Based on the results, *T. salsuginea* seemed to accumulate these soluble carbohydrates more effectively at both early and later stages of salt stress. The onset of accumulation of these soluble sugars under 100 mM [NaCl] was much more rapid and the levels far greater in *T. salsuginea* after just 12 h compared to 24 to 72 h in *A. thaliana.* The amount of these sugars was also seen to be higher in *T. salsuginea* than in *A. thaliana* under control conditions, especially for glucose. Moreover, the accumulation of glucose in *A. thaliana* was lower in control plants compared to salt-treated plants. This could be due to the plant favoring the production of fructose over glucose, with a limited amount of sucrose (1.5-fold increase) accumulated in *A. thaliana.* The sugar alcohol inositol is a ubiquitous six-carbon cyclohexane hexitol, and its derivatives pinitol and D-ononitol are implicated as osmoregulators in various biological systems [28]. In addition to this, inositol and its methylated derivatives are also implicated in various other cellular functions, like the regulation of growth, membrane biogenesis, signal transduction, and membrane dynamics [29].

There was a positive correlation between the levels of sucrose and the *SUS3* transcript in *T. salsuginea* but not in *A. thaliana*. This might suggest the importance of this gene in the modulation of carbon metabolism under salt stress in *T. salsuginea* and potentially other salt-resistant plants. The observed increase in sucrose levels in *A. thaliana* might be controlled by a different isogene of SUS or by post-transcriptional regulations of the *SUS3* enzyme in this species. This difference further highlights the complexity of the salt tolerance trait; different plant species might use different regulations of the same isogene and/or different isoforms of the same gene to respond to salt stress.

These differences in carbohydrate accumulation in both species under stress and non-stress conditions make it difficult to assign carbohydrates with the function of a primary response to salt stress, as an increase in any one of them may be a result of the reactivation of photosynthesis regulated through the onset of other defense mechanisms [30]. The various mechanisms that may regulate metabolic fluxes and signaling pathways all together make a complex network that controls the intracellular levels of these sugars. Regardless of how they accumulate and what their source might be, they do accumulate very quickly and at higher levels in *T. salsuginea* than in *A. thaliana*. This makes *T. salsuginea* potentially more effective under salt stress in terms of mobilizing sugars that may move throughout the plant, fulfilling roles as major energy sources, precursors for many metabolites, signaling components, osmoregulators, and ROS scavengers [30]. Therefore, sugars might help the plant to maintain high photosynthetic capacities in the continuous presence of salt.

In the case of organic acids, only malate was measured in the two plant species, since it has important roles in most of the plant organelles. Malate is rapidly transferred between the different subcellular compartments due to the many transport systems, and the movement of inter-organellar malate has been reported under stress conditions [31] Various biological functions involve malate, as described by [32], including (1) the control of cellular pH, (2) the support of photorespiration, (3) redox homeostasis, (4) stomatal movement via the regulation of osmotic pressure, (5) and the transport and exchange of reduced equivalents between cellular compartments. From comparing the response kinetics between the two species, two main differences surfaced. First, *A. thaliana* showed a slight increase and *T. salsuginea* showed a substantial drop in the accumulation of malate upon salt treatment. Second, under non-stress conditions, the amount of malate throughout 10 days remained at much higher levels in *T. salsuginea*. This might suggest that *T. salsuginea* is pre-programmed to tolerate salt stress, i.e., it has an exclusive feature which is commonly related to halophytes that involves constitutive and adaptive mechanisms, making it metabolically ready in the anticipation of stress [33]. And the observed malate reduction could mean that under salt stress, *T. salsuginea* favors other specialized compounds or even sugars as a carbon source. In contrast to *A. thaliana*, the channeling of energy and carbon to produce organic solutes for sustaining the high level of tolerance to salinity is deployed very early and to higher extents in *T. salsuginea*.

This key factor can also be understood from the results for the different accumulation kinetics of proline between the two species. Proline accumulates in various higher plants, and it is commonly regarded as the main effector response (with other hexoses) to salt stress and can contribute to around 50% of the osmotic adjustment [34,35]. *T. salsuginea* seems to favor proline accumulation, particularly in the early stages of salinity, and thus apparently has machinery capable of regulating such a response. In control plants, *T. salsuginea* again shows its pre-programmed characteristic of facing salt stress with high proline levels compared to *A. thaliana*. These results are in accordance with those reported by [36]. Proline accumulation after 24 h and 72 h of salt stress was 14-fold higher in *T. salsuginea* at 100 mM [NaCl]. But these differences dropped to 2 folds after 5 days and to the same level after 10 days of salt treatment between the two species. This could be due to the fact that within 24 to 72 h, *T. salsuginea* was able to activate/deactivate and increase/decrease different complex mechanisms to defend against salinity and quickly adapt to changed conditions. Increased levels of proline might assist in the acclimation to salinity by lessening the effects of salt on cell membranes, regulating the accumulation of available N protecting enzyme activities and acting as signaling/regulatory molecules to activate multiple other responses [37]. Such responses would increase the plant responsiveness to salt early in the exposure period to mount a range of required acclamatory mechanisms to achieve functional stability throughout the plant body and help the plant survive during extended periods of salinity. There was a positive correlation between transcript levels for the *P5CS1* gene and proline levels over the 10-day course of the experiment in both species. This result highlights the key role of this gene in the response to salt stress and the strong control over the accumulation of proline under stress conditions.

The results suggest a differential regulation of the accumulation of metabolites under salt stress in the two close relatives, *A. thaliana* and *T. salsuginea*. *T. salsuginea* showed faster and stronger responses to salt stress with potentially greater osmoregulation and better control over salt uptake and partitioning. These differences in the kinetics and/or amplitude of responses in *T. salsuginea* compared to *A. thaliana* were observed in the regulation of the accumulation of key compatible metabolites, such as sucrose, fructose, inositol, and proline, for enhanced stress tolerance. The two species have over 90% sequence similarity at the genome level, yet they exhibited a striking difference in salinity tolerance. Various direct comparisons between the two plant systems have been made in the past, and these have provided exciting results offering more insight. This work supports and backs up the recent emerging paradigm that the higher salt tolerance exhibited by *T. salsuginea* is a matter of the differential regulation of certain processes and demonstrates that these processes are deployed at a slower rate and to a lower extent in *A. thaliana* under salt stress compared to *T. salsuginea*. In addition to a quicker response to salt stress, *T. salsuginea* benefits from the higher constitutive level of expression of the assessed responses. These responses are constitutively expressed under unstressed conditions, which indicates a pre-adaptation to salinity in this species. At the molecular level, this preadaptation might be encrypted in the gene promoter sequence and/or in the gene sequence itself, impacting the expression of the key genes of the salt response pathways and/or the splicing and stability of their transcripts. In addition, epigenetic regulation, including via DNA methylation and histone modification, might play an important role in fine-tuning the expression of the salt responses. These regulations might be more efficient in salt-tolerant species like *T. salsuginea* [38]. Epigenetic regulations play a key role in the regulation of gene expression via the opening of DNA molecules, facilitating the access of the transcriptional machinery to the genes. Many recent reports have associated increased epigenetic regulation with salt stress responses in different plant species, including cereals [39]. These results indicate the need to deeply investigate the regulation of gene expression and transcript accumulation in the two species. The additional knowledge provided by this work about the physiological differences between *A. thaliana* and *T. salsuginea* in terms of responses to salinity can be used to inform comparisons between the two species at the genome sequence levels, since the genome sequences for the two plant species are available. Comparative analysis of the genes/transcripts and the regulatory sequences might help determine the key evolutionary mechanism behind the better adjustment/resistance to salinity exhibited by *T. salsuginea*.

## 4. Materials and Methods

### 4.1. Plant Growth and Salt Treatment

*A. thaliana* (Columbia) and *T. salsuginea* (Yukon) seeds were surface sterilized using 70% ethanol, washed three times with sterile water, and sown on John Innes soil compost No. 3. The pots were placed at 4 °C for 72 h to synchronize germination. The pots were then transferred to a controlled growth room for a 12 h photoperiod with a light intensity of 150 μmol m^−2^ s^−1^ at plant height and a thermoperiod of 23 °C at day and 18 °C at night. Seven-day-old seedlings were then transferred to separate pots containing moist John Innes soil compost No. 3. Then, 4-week-old *A. thaliana* and 6-week-old *T. salsuginea* plants, similar in size and before bolting, were divided into 3 sets and irrigated with 3 different NaCl concentrations prepared in tap water. *A. thaliana* was watered with 0, 50, and 100 mM [NaCl], and *T. salsuginea* was watered with 0, 100, and 500 mM [NaCl] (0 mM refers to tap water) at a fixed time (12:00) every day for 10 days. Shoots and roots were harvested at a fixed time (16:00) as 3 plants per sample after 12 h and 1, 3, 5, and 10 days of the salt treatment, weighed, and frozen in liquid nitrogen. Three samples were harvested at each time point for each NaCl concentration for both plant species. Control plants were watered with tap water only and harvested in parallel to salt-treated plants.

### 4.2. Determination of Growth Rate

The dry weight (Dw) of entire plants was determined after the desiccation of samples at 80 °C for 24 h and used to assess plant growth.

### 4.3. Determination of Sodium and Potassium Contents

The contents of Na^+^ and K^+^ in shoot samples were determined by the dry combustion method using hydrochloric acid [40]. Shoot samples were ground under liquid nitrogen into fine powder and weighed in ceramic crucibles. The samples were ashed overnight in a furnace at 450 °C and weighed again after incineration. Ashed samples were then dampened using a few drops of sterile water and left to evaporate in a hot steam bath after the addition of 2 mL of concentrated (11M) HCl. After drying, samples were dampened again using sterile water and left in the hot steam bath for 1 h. This was followed by the addition of 10 mL of 25% HCl, and the samples were warmed slightly to form the extract which was filtered through Whatman No. 1 filter paper into a 100 mL volumetric flask. The crucibles were washed many times using 1% HCl, and washings were transferred through the same filter paper to the volumetric flask. The extract was allowed to cool down and then made up to the volume of 100 mL with sterile distilled water. Standards for sodium and potassium measurements were prepared in parallel to the samples. The standards ranged from 0 to 25 µg, and Na^+^ and K^+^ were measured using a flame photometer after appropriate dilutions.

### 4.4. Determination of Proline Content

A colorimetric method was used for measuring proline content [41]. Powdered tissue (100 mg) was homogenized in 1 mL of 3% (*w*/*v*) aqueous sulfosalicylic acid solution. After clarification by centrifugation at 10,000× *g* for 3 min. at room temperature, the homogenate of the supernatant (500 μL) was mixed with glacial acetic acid and acidic ninhydrin reagent (500 μL each) in a new 2 mL tube. The mixture was incubated in boiling water for 1 h, and then the reaction was stopped by placing the tubes at room temperature (19 –23 °C) for 5 min. Readings at 546 nm were taken immediately using a spectrophotometer, and the concentration of proline was determined against a standard curve produced using commercial L-Proline prepared in parallel to the samples and calculated on a dry weight basis (μg proline/mg DW).

### 4.5. Analyses of Sugars and Sugar Alcohols

The levels of sucrose, fructose, glucose, and inositol in control and salt-treated plants from the two plant species were determined using high-performance ion chromatography with pulsed amperometric detection (PAD). The analyses were conducted with three biological replicates for each time point. Exactly 500 mg of plant tissue was homogenized in 5 mL of 80% methanol, followed by incubation at 75 °C for 40 min. The insoluble fraction including debris was removed from the methanol extract by centrifugation at 3500× *g* for 6 min. at room temperature. Half (2.5 mL) of the methanol extract was dried by evaporation using a sample concentrator overnight and resuspended in 1 mL of molecular-grade water. The extract was then desalted using a column of the Dowex AG50W X4–200 and Amberlite IRA–67 series. The eluate was then injected (20 μL) into an HPLC column (CarboPac PA100 with guard, Dionex, Sunnyvale, CA, USA), followed by isocratic separation with a single eluent consisting of 150 mM NaOH to separate and determine the concentration of sugars including sucrose, glucose, fructose, and inositol in each sample. Sugars were identified and quantified in the separation profiles by retention time in a column based on that of commercial standards (Sigma-Aldrich, Gillingham, UK).

### 4.6. Determination of Malate Content

Malate content was measured in salt-stressed and unstressed *A. thaliana* and *T. salsuginea* plants. Leaf tissue was homogenized in 80% methanol and incubated at 75 °C for 40 min. The homogenates were clarified by centrifugation at 10,000 rpm for 3 min at room temperature, and the supernatants were transferred to a 10 mL falcon tube and dried overnight using a sample concentrator and then resuspended in 2 mL of 200 mM bicine/KOH buffer (pH 9.5) containing 1M glycine, 0.4 M hydrazine sulphate, and 10 mM EDTA. Malate was measured following the method described by [42] based on an enzymatic assay using L-malate dehydrogenase (MDH) and NAD. The difference in OD was plotted against a calibration curve generated using commercial L-malate run in parallel to samples.

### 4.7. RNA Extraction and qRT-PCR

Total RNA was isolated from the shoots of three replicate plants using the Tri-reagent method, as described by [43]. Approximately 100 mg of ground tissue was homogenized in 1 mL of Tri-reagent (Helena Bioscience, Gateshead, UK). The samples were then incubated in the fume hood at room temperature for 10 min. Then, 250 μL of chloroform was added to them. The suspensions were then mixed by vortexing and left at room temperature for 5 min. They were then centrifuged at 13,000 rpm for 10 min at 4 °C. The upper phase was transferred to a 1.5 mL RNase/DNase-free tube, and the RNA was then precipitated in the presence of 200 mm NaCl, 133 mm Na-Citrate, and 17% (*v*/*v*) isopropanol and pelleted by centrifugation at 13,000× *g* for 30 min at 4 °C. RNA pellets were washed with 1 mL of ice-cold 70% (*v*/*v*) ethanol, air-dried for 15 to 30 min, and resuspended in 20 μL of diethyl pyrocarbonate-treated water.

To monitor changes in transcript levels for *P5CS1*, *SOS1*, and *SUS3* genes, semi-quantitative RT-PCR was used. Ubiquitin (*UBQ10*) was used as the reference gene based on early reports showing its stable transcript levels in both *A. thaliana* and *T. salsuginea* [44]. Brilliant II SYBR Green QRT-PCR master mix (Agilent Technologies, Cheadle, UK) was used according to the manufacturer’s instructions. Exactly 100 ng RNA for all the target genes and 10 ng RNA for the *UBQ10* gene with 100 nM of gene-specific primers (see sequences below) were used in 25 µL reactions. QRT-PCR was performed using the following thermal profile: 59 °C (*P5CS1*), 53 °C (*SOS1*), and 55.3 °C (*UBQ10*) for 30 min (reverse transcription), followed by denaturation at 95 °C for 2 min and 40 PCR cycles consisting of 94 °C for 15 s, 59 °C (*P5CS1*), 53 °C (*SOS1*), 55.3 °C (*UBQ10*) for 30 s, a plate read, and 72 °C for 1 min. Melting analysis was performed between 45 and 90 °C at the end of each QRT-PCR run to confirm the specificity of the amplified products. Q-RT-PCRs were run in triplicate for each sample, and each time point consisted of three samples. Mean Cts were used to calculate the relative gene transcript levels of the target genes using the standard 2^−(∆∆Ct)^ method, where the Ct of the unstressed control samples was subtracted from the Ct of the salt-treated samples and normalized against the delta Ct of the reference gene (*UBQ10*) [45].

### 4.8. Δ1-Pyrroline-5-Carboxylate Synthase 1 (P5CS1), Amplicon Size: 80 bp

Forward primer: 5′-GAGCTAGATCGTTCACGTGCTTT-3′

Reverse primer: 5′-ACAACTGCTGTCCCAACCTTAAC-3′

### 4.9. Salt Overly Sensitive 1 (SOS1), Amplicon Size: 130 bp

Forward primer: 5′-CCTTACACTGTCGCTCTTCTCGTTA-3′

Reverse primer: 5′-TTAGCTCCATATTCGAGAGATCCA-3′

### 4.10. Sucrose Synthase 3 (SUS3), Amplicon Size: 188 bp

Forward primer: 5′-GACCAAGACCTGGTGTTTGGG-3′

Reverse primer: 5′-AGACGAACGAGAAGGACGTGG-3′

### 4.11. Ubiquitin 10 (UBQ10), Amplicon Size: 58 bp

Forward primer: 5′-CTCTCTACCGTGATCAAGATGCA-3′

Reverse primer: 5′-TGATTGTCTTTCCGGTGAGAGTC-3′

### 4.12. Sampling and Statistical Analysis

Three samples were taken at each time point, and each sample consisted of three plants ground together. To limit variability, ensure reproducibility, and limit the effect of the natural circadian changes, all plants were strictly watered at the same time (12:00), and samples were taken at the same time (16:00) and in the same way.

The normality of distribution of the data was verified in Excel, and an independent T test was performed to determine the significance of differences between *A. thaliana* and *T. salsuginea* using SSP statistical software. For broader comparisons of the responses between groups over the full-time course of the analysis, an ANOVA test would have been more suitable, but this was not conducted in this study. Nevertheless, the T-test results were robust enough to allow for evidenced conclusions.

The correlation between the profile of changes in metabolites and transcript levels under the salt stress effect across the 10-day course of the experiment was calculated as % using the Excel correlation function.

## 5. Conclusions

Large variation in salt tolerance exists among plant species and even between genotypes of the same species. Understanding the mechanisms underpinning this variation is of paramount importance for engineering higher salinity tolerance in key crops. The current study compared the kinetics of responses to salinity in terms of growth and osmoregulation between two close relatives, *A. thaliana* and *T. salsuginea*. The main findings show the following.

### 5.1. Potassium

Salt stress induced in *T. salsuginea* initially a rapid increase in K^+^ uptake, followed by a decrease after 2 days of exposure to salt. In contrast, a steady decrease was observed in *A. thaliana* from the start of salt stress. This result confirms the possible contributing role of K^+^ in the higher salt resistance exhibited by *T. salsuginea*.

### 5.2. Osmoregulation

*T. salsuginea* responded more rapidly and at higher levels to salt stress than *A. thaliana* in terms of osmotic adjustment by increasing the concentration of proline, soluble sugars, and inositol. The constitutive levels of sucrose, glucose, inositol, proline, and malate were higher in *T. salsuginea* compared to *A. thaliana*, suggesting that *T. sulsuginea* has higher metabolic plasticity in responding to osmotic stress, consequent to salt exposure.

### 5.3. Organic Acids

The levels of organic acids decreased under salt stress in *T. salsuginea*, while they increased in *A. thaliana*. This contrasting response might suggest a distinctive carbon allocation response to salt stress between the two species. In *T. salsuginea*, organic acids might have been broken down from a relatively high concentration to feed the higher metabolic response for the osmotic adjustment to salt stress.

### 5.4. The Kinetics of the Salt Response

Although the two species seem to engage the same metabolic pathways to respond to salt stress, the rapidity and the level of the response, together with the direction (increase or decrease) of the response, seem to represent a credible explanation of the differential salt tolerance exhibited by the two close relatives. These differences are very likely to be controlled by differences in the control of gene expression. Variation in the promoter regions of key salt resistance genes and/or in transcription factors might be the ultimate source of the observed metabolic differences and consequently the variation in salt tolerance.

## Figures and Tables

**Figure 1 ijms-26-05141-f001:**
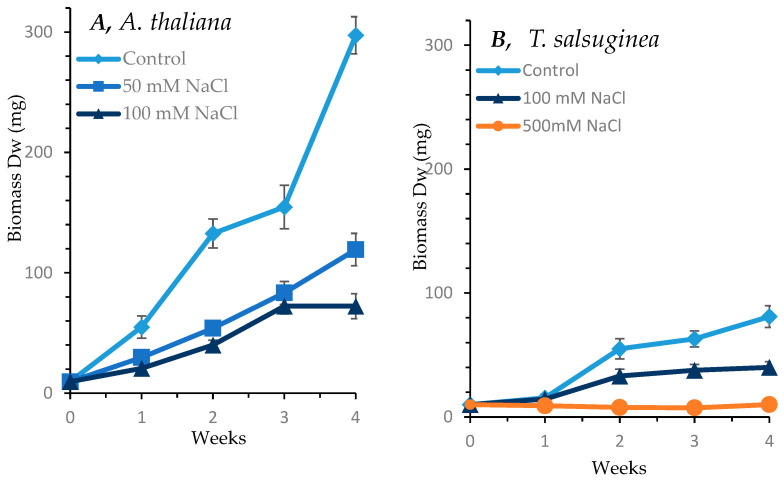
Effect of salt stress on shoot growth over a period of 4 weeks in *A. thaliana* (**A**) and *T. salsuginea* (**B**), expressed as dry weight (DW). Each point is a mean of three replicates, and error bars are standard errors calculated from the three replicates (*p* ≤ 0.002 at 10 days of salt stress at 100 mM).

**Figure 2 ijms-26-05141-f002:**
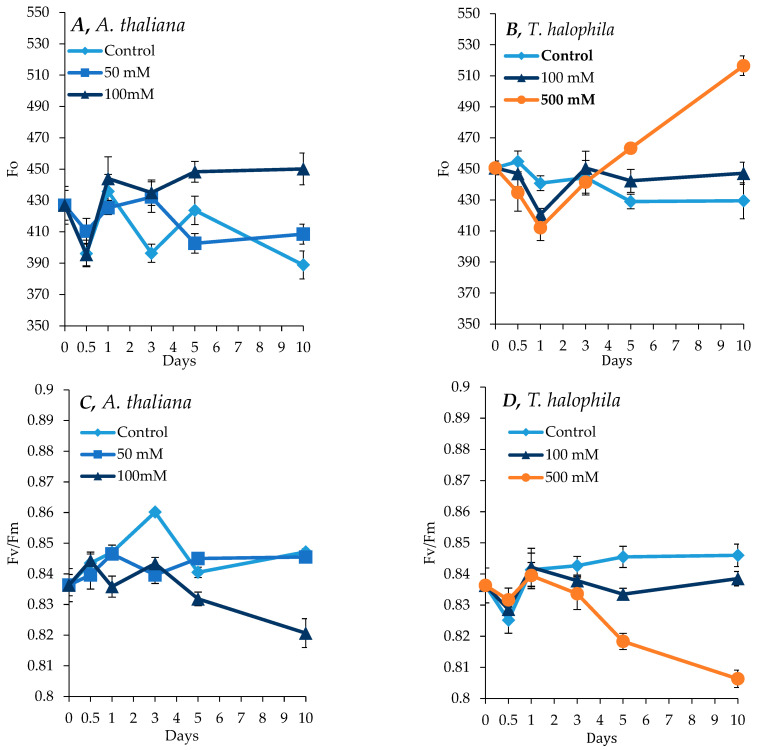
Maximum chlorophyll fluorescence (Fo) and PSII activity (Fv/Fm) in *A. thaliana* (**A**,**C**) and *T. salsuginea* (**B**,**D**) plants subjected to salt stress. Each point is a mean of three replicates, and error bars are standard errors calculated from the three replicates (*p* ≤ 0.012, at 10 days of salt stress at 100 mM).

**Figure 3 ijms-26-05141-f003:**
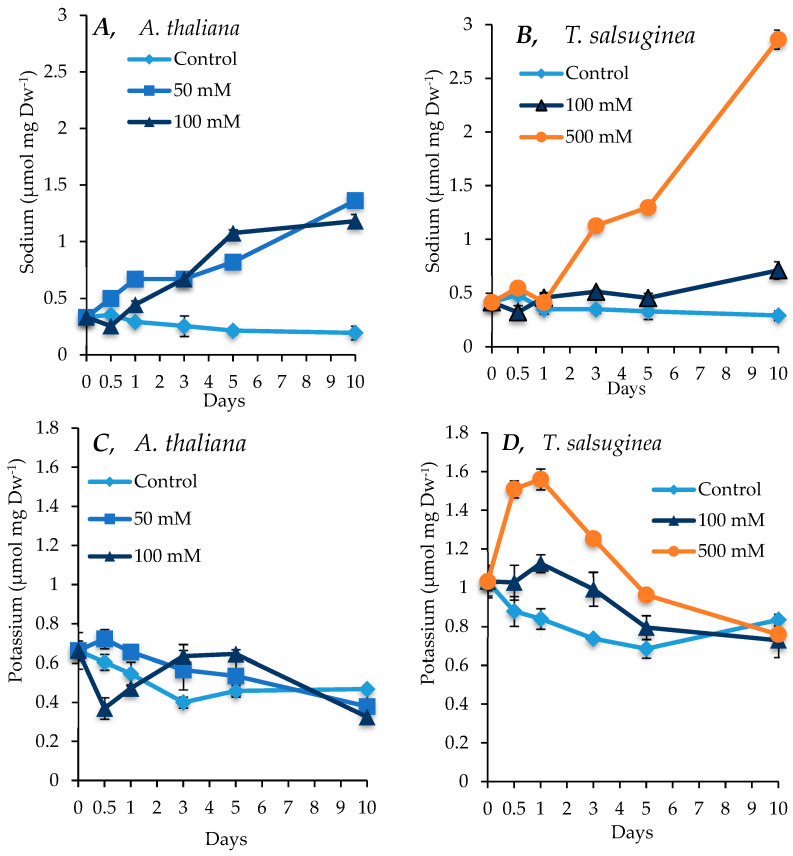
Levels of Na^+^ and K^+^ measured in shoots of *A. thaliana* (**A**,**C**) and *T. salsuginea* (**B**,**D**) over a 10-day period of salt treatment (NaCl). Each point is a mean of three replicates, and error bars are standard errors calculated from the three replicates (*p* ≤ 0.02, at 10 days of salt stress at 100 mM).

**Figure 4 ijms-26-05141-f004:**
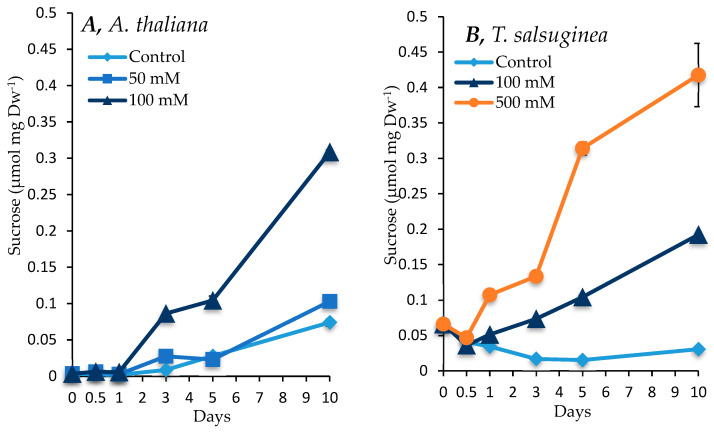
Levels of sucrose measured in shoots of *A. thaliana* (**A**) and *T. salsuginea* (**B**) over a 10-day period of salt treatment (NaCl). Each point is a mean of three replicates, and error bars are standard errors calculated from the three replicates (*p* < 0.001, at 10 days of salt stress at 100 mM).

**Figure 5 ijms-26-05141-f005:**
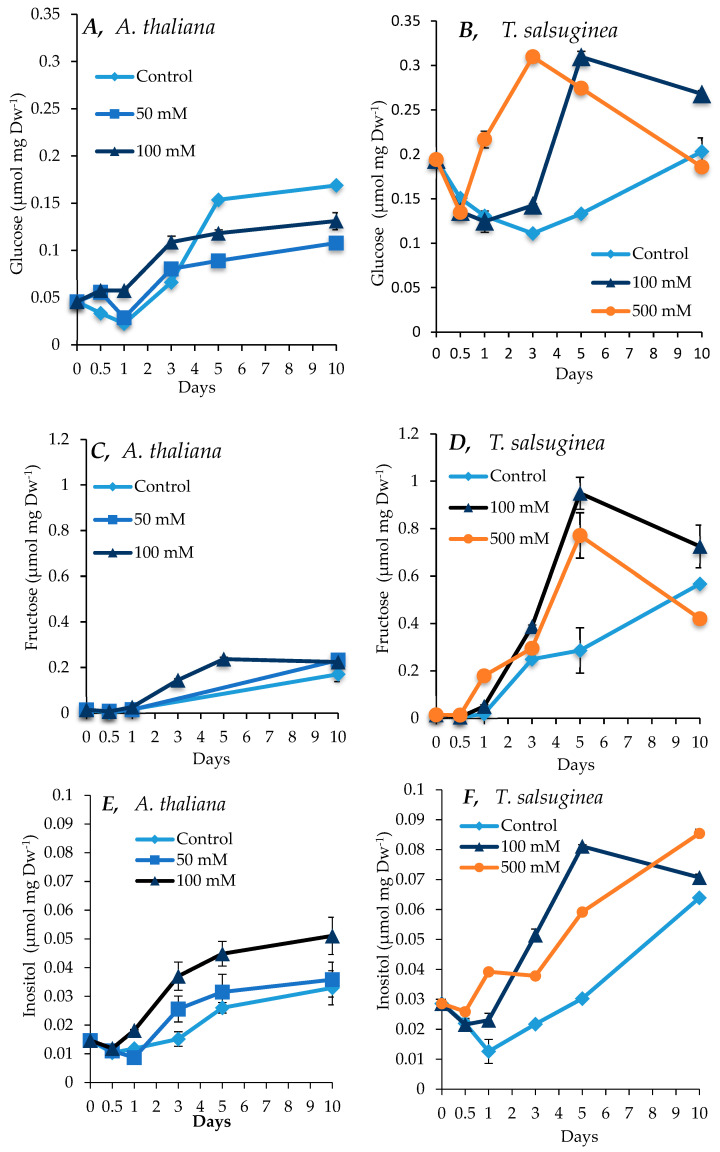
Levels of fructose, glucose, and inositol measured in shoots of *A. thaliana* (**A**,**C**,**E**) and *T. salsuginea* (**B**,**D**,**F**) over a 10-day period of salt treatment (NaCl). Each point is a mean of three replicates, and error bars are standard errors calculated from the three replicates (*p* < 0.002, at 10 days of salt stress at 100 mM).

**Figure 6 ijms-26-05141-f006:**
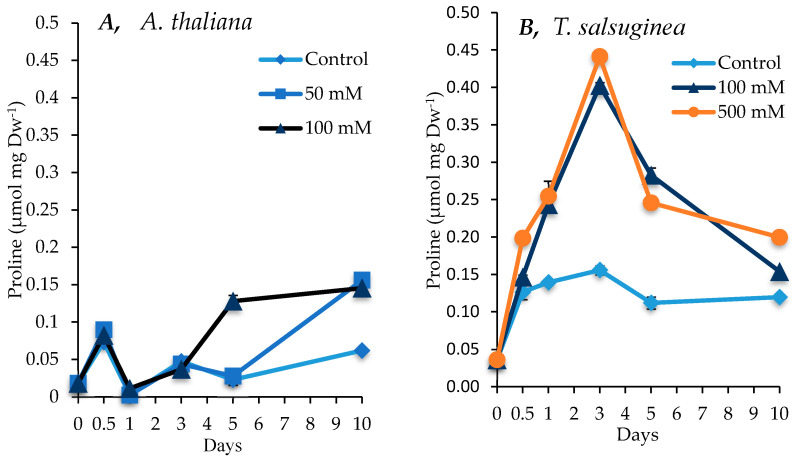
Level of proline measured in shoots of *A. thaliana* (**A**) and *T. salsuginea* (**B**) over the 10-day period of salt treatment (NaCl). Each point is a mean of three replicates, and error bars are standard errors calculated from the three replicates (*p* < 0.005 at 3 and 5 days of salt stress at 100 mM).

**Figure 7 ijms-26-05141-f007:**
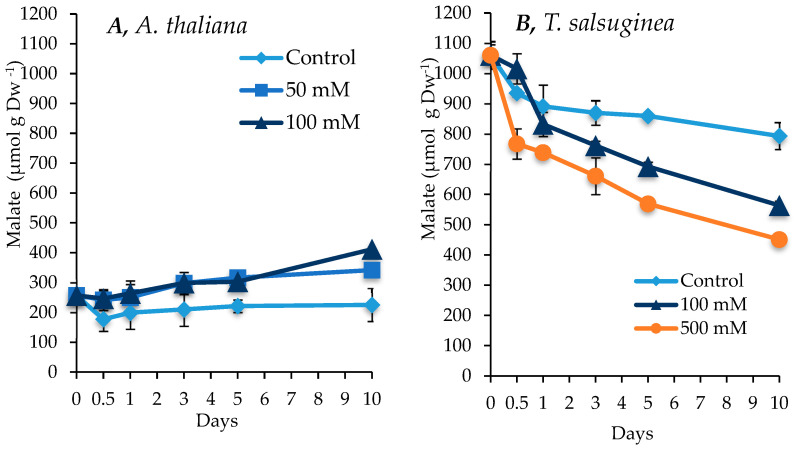
Change in malic acid levels in shoots of *A. thaliana* (**A**) and *T. salsuginea* (**B**) induced by salt treatment (NaCl) over 10 days. Each point is a mean of three replicates, and error bars are standard errors calculated from the three replicates (*p* < 0.005 at 10 days of salt stress at 100 mM).

**Figure 8 ijms-26-05141-f008:**
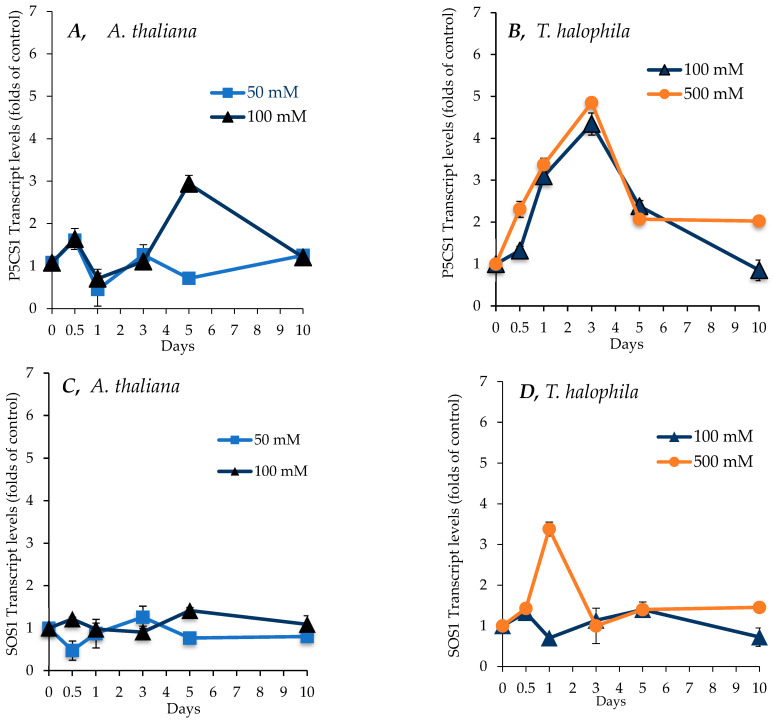
Transcript levels for a *P5CS1* gene encoding Δ1-Pyrroline-5-carboxylate synthase 1, a key enzyme for proline synthesis, an *SOS1* gene encoding a Na^+^/H^+^ antiporter, and an *SUS3* gene encoding sucrose synthase, which was responsible for breaking down sucrose under salt treatment (NaCl) over 10 days in shoots of *A. thaliana* (**A**,**C**,**E**) and *T. salsuginea* (**B**,**D**,**F**). Transcript levels are reported as folds of unstressed control plants. Each point is a mean of three replicates, and error bars are standard errors calculated from the three replicates (*p* < 0.005 at 3 and 5 days of salt stress at 100 mM).

**Table 1 ijms-26-05141-t001:** Correlation shown as (%) change in metabolites, proline, sucrose, and transcript levels for *P5CS1* and *SUS3* genes, respectively, in *A. thaliana* and *T. salsuginea* subjected to salt stress at 100 mM [NaCl] over a 10-day course.

Metabolite/Gene	Correlation (%)	
	*A. thaliana*	*T. salsuginea*
Proline/*P5CS1*	66	87
Sucrose/*SUS3*	−32	89

## Data Availability

The datasets used and/or analyzed during the current study are available from the corresponding author on reasonable request.

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
