# Peer review of "Differences in the Temporal Kinetics of the Metabolic Responses to Salinity Between the Salt-Tolerant Thellungiella salsuginea and the Salt-Sensitive Arabidopsis thaliana Reveal New Insights in Salt Tolerance Mechanisms"

_ijms, 2025, doi:10.3390/ijms26115141_

Round 1
Reviewer 1 Report
Comments and Suggestions for Authors
This manuscript provides a comprehensive comparative analysis of the temporal metabolic, physiological, and transcriptional responses to salinity in Arabidopsis thaliana (Columbia) and Thellungiella salsuginea (Yukon). By combining detailed time-course sampling with biochemical, physiological, and gene expression analyses, the study reveals critical differences in how these species manage salinity stress.
Key strengths include:
- A well-structured temporal framework capturing responses from 12 hours to 10 days.
- Multiple levels of analysis: growth performance, chlorophyll fluorescence, ion homeostasis, osmolyte accumulation, and transcriptomics.
- Species-specific salt treatments aligned with differential tolerance levels.
The study delivers valuable insights into salt tolerance mechanisms, particularly the importance of anticipatory metabolic states and rapid response deployment in T. salsuginea. However, several points must be addressed to enhance clarity, scientific rigor, and interpretive depth.
Major Comments:
- Statistical Approach:
The data were assessed for normality and analyzed using independent T-tests. Given the repeated measurements over time, the use of repeated measures ANOVA or mixed-effects modeling would enhance the robustness of the conclusions. At a minimum, this should be acknowledged as a limitation.
Clarify in the methods whether multiple comparisons corrections were applied and report exact p-values or significance thresholds in figure legends or main text.
Physiological Relevance and Mechanistic Insight:
The hypothesis that T. salsuginea exhibits “pre-adapted” metabolic states under control conditions is well-supported and aligns with the observed higher basal metabolite levels. This concept of metabolic preparedness or priming should be better developed and linked to evolutionary or ecological adaptations to saline environments.
Consider discussing potential roles of epigenetic regulation or stress memory in enabling rapid metabolic and transcriptional activation.
I suggest including a graphical summary or model depicting the sequence of key metabolic and transcriptional events in both species under salinity.
Consider a supplementary table correlating metabolite levels and gene expression data across time points.
Where possible, briefly speculate on whether the early responses in T. salsuginea are primarily due to transcriptional pre-activation, metabolite reserves, or faster signal transduction.
Minor Comments
- Improve language precision: e.g., “responded quicker” → “responded more rapidly”; “background levels” → “constitutive levels”.
- Ensure consistency in naming (e.g., Thellungiella salsuginea T. salsuginea).
- All figure legends should specify that error bars represent standard error (SE) and note the number of biological replicates.
- In methods, ensure sufficient detail on HPAEC-PAD setup, proline assay, and RNA extraction to support reproducibility.
- The time of day used for watering (12:00) and sampling (16:00) is a strength and supports reproducibility, highlight this as part of methodological rigor.
Reviewer 2 Report
Comments and Suggestions for Authors
In this study, Aayush Sharma and Tahar Taybi compaired the differences in the salt-tolerant, Thellungiella salsuginea and the salt-sensitive Arabidopsis thaliana. They performed many experiments to measure several physiological indicators and also, detected the expression profiles of 3 related genes.
I think the results are interesting and in line with expectations. However, I still have some suggestions that I hope the authors will accept.
- I'm quite dissatisfied with the format of the figures in this study. They lack consistency. For example, the font of the legends is bold in some cases but not in others. Moreover, the font format of the figures is inconsistent and rather casual.
- The types of graphs used in this study are rather monotonous, as they are all line graphs. I suggest that the authors appropriately use some other formats.
- The colors of the figures in this study are very plain. It is recommended to use a more colorful palette.
- The triangles, squares, and circles in the figures are too small.
- All data results must be subjected to a significance test for differences.
Round 2
Reviewer 1 Report
Comments and Suggestions for Authors
Agree with the revised manuscript.